# Proposal for Tower Crane Productivity Indicators Based on Data Analysis in the Era of Construction 4.0

**Thomas Danel** [1,*], **Zoubeir Lafhaj** [1], **Anand Puppala** [2], **Sophie Lienard** [3] **and Philippe Richard** [3]

1   CNRS, LAMcube—Laboratoire de Mécanique, University of Lille, Centrale Lille, UMR 9013, Multiphysique, Multiéchelle, 59000 Lille, France; zoubeir.lafhaj@centralelille.fr
2   Zachry Department of Civil and Environmental Engineering, Texas A&M University, College Station, TX 77843-3136, USA; anandp@tamu.edu
3   Bouygues Construction, Challenger, 1, Avenue Eugène Freyssinet, Guyancourt, 78061 Saint-Quentin-En-Yvelines, France; S.LIENARD@bouygues-construction.com (S.L.); p.richard@bouygues-construction.com (P.R.)
*   Correspondence: thomas.danel@centralelille.fr

**Abstract:** This article proposes a methodology to measure the productivity of a construction site through the analysis of tower crane data. These data were obtained from a data logger that records a time series of spatial and load data from the lifting machine during the structural phase of a construction project. The first step was data collection, followed by preparation, which consisted of formatting and cleaning the dataset. Then, a visualization step identified which data was the most meaningful for the practitioners. From that, the activity of the tower crane was measured by extracting effective lifting operations using the load signal essentially. Having used such a sampling technique allows statistical analysis on the duration, load, and curvilinear distance of every extracted lifting operation. The build statistical distribution and indicators were finally used to compare construction site productivity.

**Keywords:** tower crane; productivity; construction site; datalogger





## 1. Introduction

According to the Cambridge Dictionary, "productivity is the rate at which a company or country makes goods, usually judged in connection with the number of people and the amount of material necessary to produce the goods." The limited data available for the construction industry suggests that construction productivity has been declining for several decades [1–3]. But the construction industry is fragmented: building houses, apartments, and offices are not the same as digging tunnels or building bridges. The question is then, are these branches all suffering from this low productivity rate, or is one branch suffering more than the others? In the construction sector, productivity is twofold: on one side, it concerns the construction company and its organizational performance, on the other side, it concerns the construction project's management on the construction site [4]. This article deals with the latter. On-site construction productivity is related to the total factor productivity taking into account the usage of the equipment, the working crew's efficiency [5], the material use [6,7], capital and energy [6]. But the lack of data makes it impossible to calculate this total factor productivity [8].

Many attempts to manage the productivity calculus has been made through decades. Among them, we can cite the measured mile approach [9,10], where many techniques of motion and time study such as work or activity sampling [11,12], stopwatch study [13], photography, videotaping, time-lapse video [14,15], questionnaire surveys or even automated productivity measurement system [16] are used. Another technique consists of measuring a partial productivity rate called apparent productivity, which relates productivity to a single type of resource, such as labor [17]. For example, apparent labor productivity is

measured by relating the installed quantity (value added to the building process) to the number of hours worked (labor input) [18]. Even for this method of calculation, biases exist, and summarizing construction productivity as average labor productivity does not represent the true picture [19,20].

All the productivity measurement approaches developed since the last decades suffer from these biases and have tried to avoid them. Now there are plenty of digital tools, such as tablets, smartphones, wearable devices, and smartwatches, that are able to track the working hours of the workers on a construction site [21], so that the productivity measurement methods can be more easily applied. Calculating labor productivity is still a quite complex task since the general contractor often delegates the finishing trades and occasionally a part of the structural phase to their sub-contractors. These subcontractors do not have to be there at a certain time on the site. They are paid to do one task before a specific deadline (fixed price basis), so even if the productivity of the construction site is linked to the subcontractor's productivity, this latter is neither calculated nor manageable. Moreover, most of the time, and for their own workers, labor hours are still manually filled in by the supervisors, which could induce errors in the time breakdown of the activities such as human perception bias, asynchrony, and rounding off during manual reporting.

This article proposes to calculate the apparent productivity of the construction site through the analysis of the data collected from tower crane activities. This is by definition only during the structural phase, i.e., when the crane is installed and effectively used. The proposed method reduces one of the biases described above: human perception. In this method, the apparent labor productivity is loosely connected to the crane operator efficiency, which is also implicitly linked to the work done underneath the crane hook, moving formwork for installation or concrete for pouring, to complement the work done by the workers.

After four industrial revolutions, the construction industry has been behind the other industries. The beginning of this decade saw a new concept coming into the industry: The Industry 4.0. The pillars of this concept are big data and analytics, autonomous robots, simulation, system integration, internet of things, cybersecurity and cyber-physical systems, cloud, additive manufacturing, and augmented reality. All of these axes can be applied to the construction industry. To help with this, an ontology was previously proposed [22] to use digital tools for the resource management of a construction site. Construction professionals now work together with start-ups and digital companies to bring the Industry 4.0 concepts, digital tools, and new ways of management into the construction industry; this is named Construction 4.0 [23]. Many technological implementations of Construction 4.0 concepts have been published over recent years, and now even the workers are 4.0 [24]. We can cite the use of cameras to monitor the worker activities [25–27] or construction progress, augmented reality tools to enhance the visualization [28], or robots for building construction [29–32].

Due to the consequences of this modernization of the construction industry and a growing maturity level in smart sensors and improved connectivity, tower cranes are now connected through a data logging system. The work proposed in this study involves the use of the tower crane data extracted from the datalogger to automatically track materials and work conducted on the construction site, which is expected to increase the productivity on-site [33].

## 2. Materials and Methods

### 2.1. Construction Project

The research work was mainly based on data from a construction project called "Conservatoire Darius Milhaud". This pilot project is the first testing ground for the exploratory work proposed in this study on construction data. The pilot project is a building site of a municipal conservatory for arts spanning 2400 square meters and offers numerous workspaces (recording studios, multi-purpose rooms, dance studios, dance/dramatic art studio) and performance spaces (auditorium).

*2.2. Tower Crane*

This construction project used a flat top tower crane, a Liebherr 202-ECB Litronic, capable of lifting a maximum weight of 10 Tons and reaching a target at 45 m, in the horizontal direction, and 50 m, in the vertical direction, from the base of the mast.

*2.3. Datalogger*

The crane was equipped with an anti-collision system in accordance with article R. 4323-28 of the French Labor Code, which says that when two or more pieces of equipment used for lifting non-guided loads are installed or mounted in a workplace in such a way that their fields of action overlap, measures shall be taken to avoid collisions between the loads or with parts of the work equipment itself. This explicitly relates to jib/jib, jib/counter-jib, over-flight of prohibited areas, and restricted areas. A restricted area can also be a zone with public places, living areas, railways, roads. It is well known that the crane's location on a construction site is therefore subject to at least one of these kinds of interferences. The consequence is that most of the cranes have to be equipped with one of these systems, especially when oral instructions to the operator are not sufficient.

Collecting the data from the lifting machine is a great opportunity for scientific research since the tower crane improves productivity on-site by lifting heavier components and moving them faster than using manual labor. The crane visualizes all the stages of value creation of the structural phase and is now an essential part of modern construction sites. Moreover, data coming directly from the machine are raw, structured, and untouched, which adds value compared to some biased data entered by a human. The feasibility of using such a system to monitor the lifting equipment in support of project control has been demonstrated before [34].

In our case, the data logging system, or datalogger, is only the recording part of an anti-collision system with a load measurement unit. It then records the 3D position of the crane hook through slewing angle, trolleying distance, and hoisting height measurements and the load in the time reference frame. The datalogger is composed of several elements to capture (sensors), process (central unit), display (display screen), and transmit information (radio link for anti-collision, 3G/4G link for datalogger). In this way, it acts as an interface between the mechanical crane and the data.

## 3. Research Methodology

*3.1. Research Objective*

The main research objective of this article is to determine the feasibility of using data collected from tower crane to measure on-site productivity.

*3.2. Research Method*

The results of this article are part of a larger research project that aims to develop digital tools for managing a connected construction site conducted in collaboration with a French construction company. The method is based on data extraction and analysis to link the activities of the crane to the productivity rates of the construction site. The following steps were used.

3.2.1. Data Collection

The collection of tower crane's raw data can be performed either by manually exporting the raw data via a web portal hosted by the anti-collision system provider or by automatically exporting the data via the construction company's database.

The existing structure of the recorded crane data consists of:

-　Complete calendar date followed by the hour, minutes, and seconds (ISO 8601);
-　Slewing angle in degrees;
-　Trolleying distance in meters;
-　Hoisting height in meters;
-　Load in tons.

The date is the basis of a data logging system as all data points are time-referenced, which gives the sampling frequency of the system. This particular system periodically collected data on whether any of the measured values reached threshold values. If the threshold for at least one of the sensors is triggered, the system records the data. These thresholds are linked to the technical specifications of the datalogger product. The slewing angle of the boom relative to the mast is a relative measure for which the origin is not known. The minimum trigger threshold for the orientation measurement is one degree. The trolleying distance of the trolley relative to the boom is an absolute measurement calculated from the increment of the trolley winch. The trigger threshold for this measurement is 50 cm.

The lifting height is an indirect measure of the length of the cable wound on the lifting winch. This height corresponds to the height of the hook at the moment of the measurement. The trigger threshold is the same as the distribution threshold: fifty centimeters.

By combining all the thresholds of slewing angle, trolleying distance, and lifting height measurement, the confidence interval of the hook position in space forms a centered ellipsoid with a depth of 0.5 m, a width of 0.8 (1 to 45 m), and a height of 0.5 m. This accuracy is not as good as some other technologies available on the market and studied in the literature such as GPS [35–37], UWB localization [38,39], or inertial measurement unit [40,41], but it is cost-effective.

The load is measured in Kilonewton (kN). The trigger threshold is 0.3 kN, i.e., just over 30 kg. The gravity will be considered equal to 10 m/s$^2$. The maximum permissible load of the crane is 10 tons.

### 3.2.2. Data Preparation

Manual exports generate comma-separated values (CSV) files, in which each line is a data record. Every value is stored as plain text, whereas it exists, for example, "datetime" or "numeric" type bringing the true meaning of the stored value. A formatting step is then needed to translate text into values.

### 3.2.3. Data Sampling

Because non-human operations of the tower crane exist, sampling the dataset is a needed step to delete the number of data records that are not interesting. Indeed, the data logging system stores the measurement of every sensor every time, even in the absence of the operator. It is possible to extract this time interval because when the operator leaves the cabin, he activates the "weather vane" mode of operation. This is an automatic system enabling the crane to act as a weather vane to avoid mechanical deformations or, even worse, the overturning of the lifting machine [42,43]. The "weather vane" data represent 17% of the total dataset in terms of observations. In terms of time, the theoretical percentage is around 63%. Indeed, one crane operator drives its crane for a full business week, which represents 48 h per week, i.e., 27% of 7 days of 24 h of recorded data. The difference between the percentages is explainable by the fact that the sampling frequency is higher when the operator is driving: the more variations appear in the signals, the more the thresholds are triggered, the more the datalogger is recording data. If no "weather vane" system had been implemented on the crane, no recordings could be done during the nights and weekends because no movements could not have occurred.

The second step of sampling is the extraction of effective lifting operations. The segmentation method first consisted of detecting crenels in the load signal, which correspond to when an object is picked up by the crane. Then the crenel is extended to consider the approaching phase (last empty moving of the hook towards the current take). A method to isolate individual crane operations can be found in the literature [34] and will be discussed in this article later.

### 3.2.4. Data Processing

Once the datasets were prepared, metrics of each operation such as weight, duration, and distance could be computed. For each operation, the maximum weight can provide the type of object the crane is handling. For example, one particular configuration of formwork or a prefabricated component can have a recognizable weight among the others. The segmentation method gives the pick-up and the drop-off time of each lifted object. The duration of the lifting operation is then deduced from the calculus: drop-off time − pick-up time = operation duration. The distances traveled by the hook can also be calculated during these intervals: Euclidean distance between the pick-up and the drop-off, or a more representative distance such as the 2D curvilinear abscissa can be calculated.

### 3.2.5. Metrics Analysis

Each metric of the operations assists in analyzing part of all the tasks realized by the crane. The distribution of weight, durations, and distances were then examined.

For the weight part, the analysis provided the type of load handled by the crane. Thereby, if the supervisor realizes that the crane manipulates a lot of small pieces such as steel props too often, he can act on his crew to focus on higher load such as formworks or concrete pouring. It could also focus more attention to fill more the props baskets or concrete skips.

Durations of the operations can be compared to the expected "activities of the crane". This document was prepared by the Methods Department to calculate the theoretical use percentage of the lifting machine as a function of the level of the building. Each activity is linked with the duration of the particular task (slab concrete pouring, unloading of the delivery truck, rebar supply). These expected durations are calculated with the mode of operations and the productivity ratios coming from periodic work time measurements on-site. The durations are then considered as the standard time for the tasks, not a desirable minimum time. Deviations from this standard can be interesting for the site manager.

As previously discussed, distances can be calculated in many ways. Euclidean distance between the pick-up and the drop-off point appears not to be sufficient to appreciate the real movement of any object, according to site supervisors. Curvilinear abscissa was computed to extract the 2D distance of the actual trajectory.

### 3.3. Software Development

The processing and analysis of crane data were carried out with the R language (version 3.6.1), a programming language that offers a free and open-source platform for data analysis, statistics, visualization, spatial analysis, mapping, scientific communication, and dashboard creation.

## 4. Results

### 4.1. Raw Data Visualization

Figure 1 gives an overview of the raw data recorded by the datalogger during a full business day. The data have undergone the first two steps detailed in the "Methodology" section and the deleting of "weather vane" data. X (color red) and Y (color blue) coordinates come from the transformation of the cylindrical coordinate system (slewing angle and trolleying distance) into a Cartesian coordinate system. Z (color green) indicates the lifting height. Finally, the load (color black) shows the load measured over time. According to the construction site managers, the only data that is interesting to see here in the raw format is the load. It is then easy to know and understand that the crane has taken one object, transported it, and dropped it off.

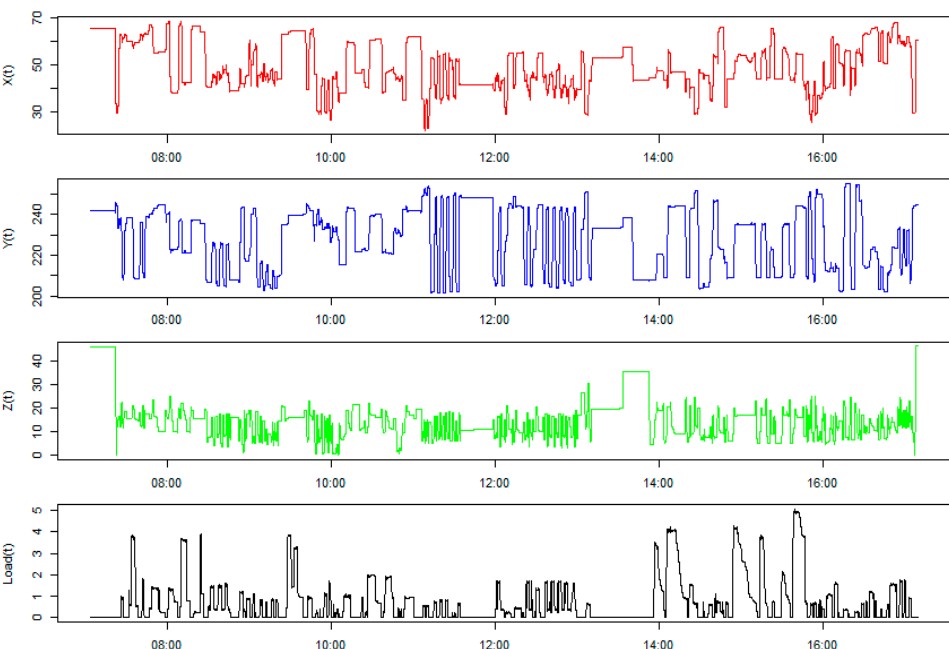

**Figure 1.** Raw data visualization over time.

Figure 2 shows a plane projection of the crane hook's positions during a full business day. The position points are colored with a color gradient as a function of the weight at the time of measurement. Date and time give the chronology of measures allowing us to link time with the crane's movements. This representation gives us a kind of heat map indicating the displacement of objects on the job site. It can be observed that more than a quarter of the accessible crane's workspace was used. Apart from that, it is difficult to say more about the movement of objects: origins, destinations, and trajectories are not visible here.

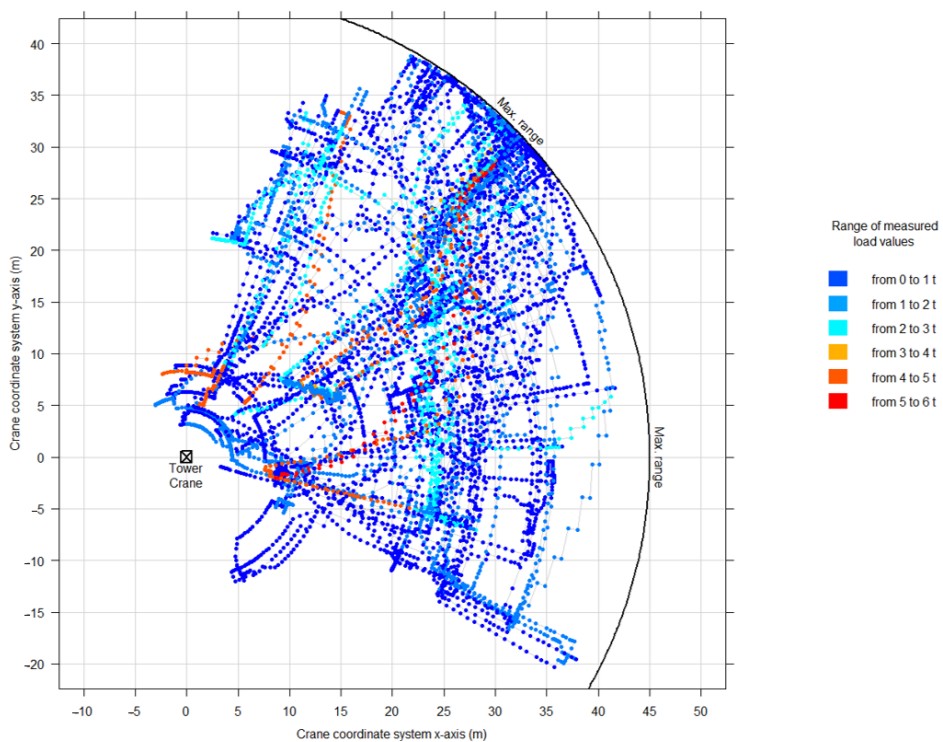

**Figure 2.** Plane projection of the crane hook's positions.

### 4.2. Crane Data Segmentation

Figure 3 shows the approach for extracting the whole, effective lifting operations comparatively using the method proposed by Sacks et al., [12]. In the current study, the empty motion just before the effective lifting operation is considered as an "Approach" phase to pick up the load. Later comes the "Attach" phase, in which the object is hooked at the crane through slings. It is then possible to "Load" it by lifting the hook, which will increase the load measured by the sensor. The "Transport" of the load is done through a combination of boom orienting and trolleying. When the object is vertically aligned with its final destination is when the "Unload" phase begins, which is done by lowering the object and dropping it on the ground. The work of the tower crane is done, and workers only need to "Detach" the object to do something with it.

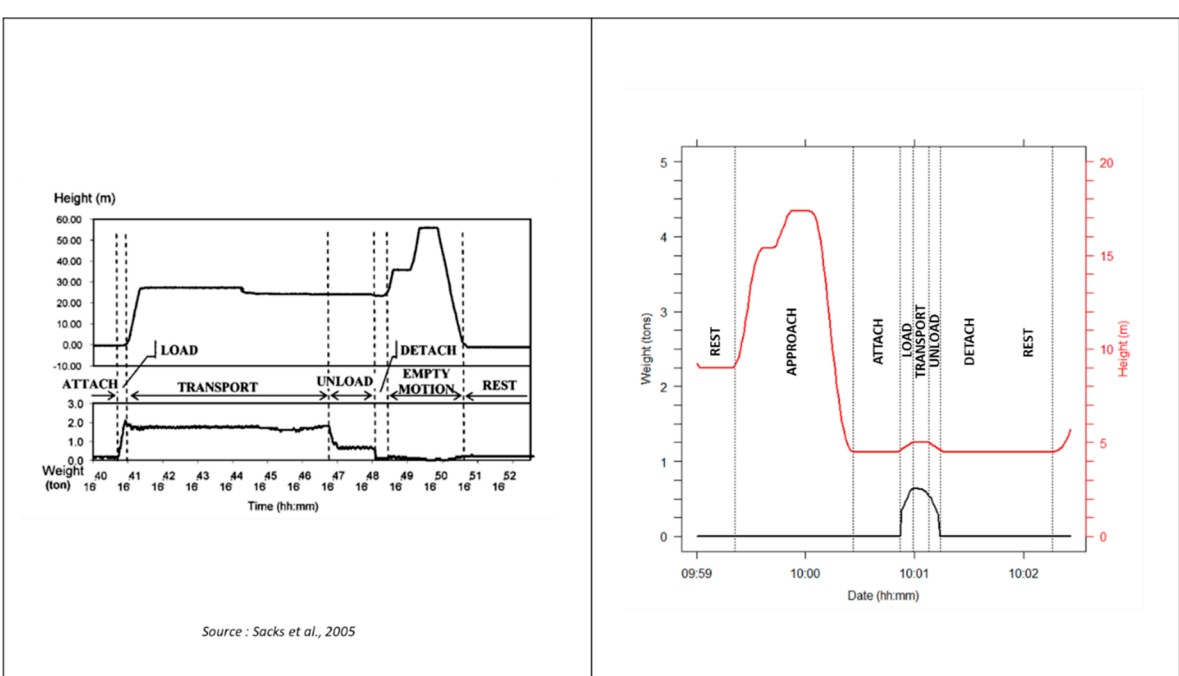

**Figure 3.** Comparative point of view between (Sacks et al., 2005) [34] and ours.

In the attaching and detaching phase, it is not possible to determine whether the object is actually attached, detached or when the crane is just idle. It happened that workers on-site took some time to arrive at the location of the object to sling it, or they have to hook the four slings instead of only two for the concrete skip, for example. In fact, each phase (Approach, Attach, Load, Transport, Unload, and Detach) could have more or less waiting time. These waiting periods can be prevented by good coordination between the crane and the construction site crew. The hypothesis in this article is that this wasted time is quite limited during the transport phase of the object. That is why the next part focuses on extracting the effective lifting operation when the measured weight is not zero.

As presented in Figure 4, data are segmented by weight to obtain actual lifting operations. The weight signal is exploited through crenels (grey rectangle) representing the presence of a load hanged to the hook (in orange). These crenels represent then the (Load–Transport–Unload) sequence. This creates temporal intervals that could be used further to identify objects by their weight or the way they are moved. The segmentation gives us then the position where the object (building component, concrete skip, or formwork, for example) was picked up then dropped off.

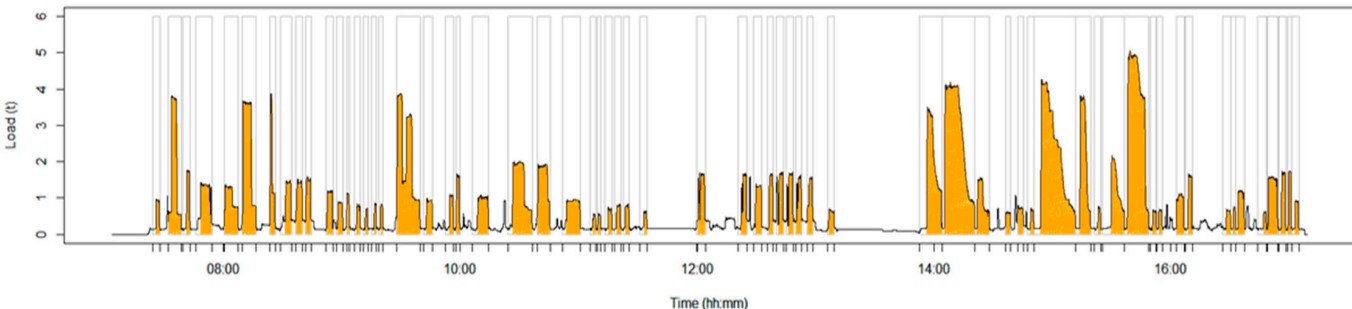

**Figure 4.** Segmentation visualization of load signal.

Figure 5 represents the position of the hook projected on a plan. Red squares are pick-up positions, whereas green triangles are drop-off positions. For this particular day, most of the positions where objects were picked up near the maximum range of the tower crane. This situation is conducive to a slowing down of the pick-up phase. The theoretical load curve given by the crane manufacturer, showing that the maximum permissible load decreased with the trolleying distance. For this crane, the lifting speed was 35 m per minute (i.e., 30.4% of the maximum lifting speed of 115 m per minute) at the maximum trolleying distance (45 m). In this case, it was better to avoid picking up objects close to the crane's maximum reach. In any case, an indicator can be traced back to the method department to indicate a non-optimal crane position.

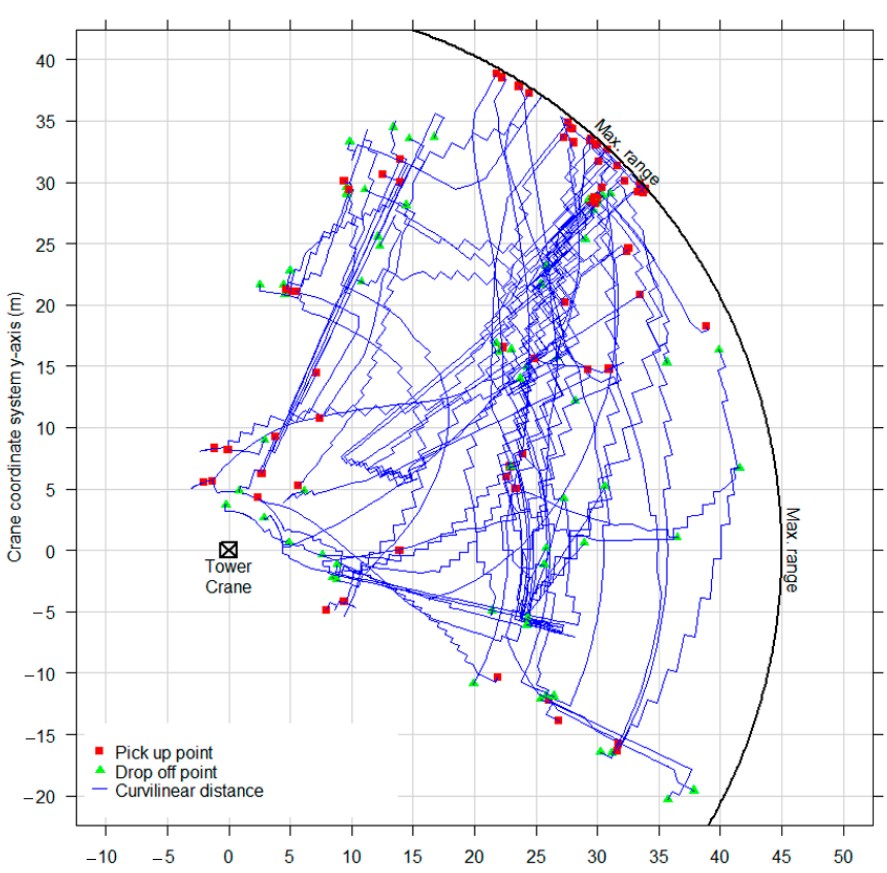

**Figure 5.** Position of the hook projected on a plan.

*4.3. Statistic Distribution of Metrics*

The dataset is composed of 1,939,662 observations representing data from 21 June 2018 to 29 May 2019. The data segmentation transformed these data into 14,848 lifting

operations. These data allow performing statistical analysis of the dataset by guaranteeing the number of samples.

Figure 6 depicts the distribution of the durations of lifting operations. The mean value is 4 min and 11 s, a shorter time compared to the mean value of 7 min determined by the Methods Department. 78% of every operation took less than the mean value. Delicate operations such as laying self-supporting slabs or installing prefabricated stairs theoretically needed 12 min, but the maximum duration measured value was far more than this.

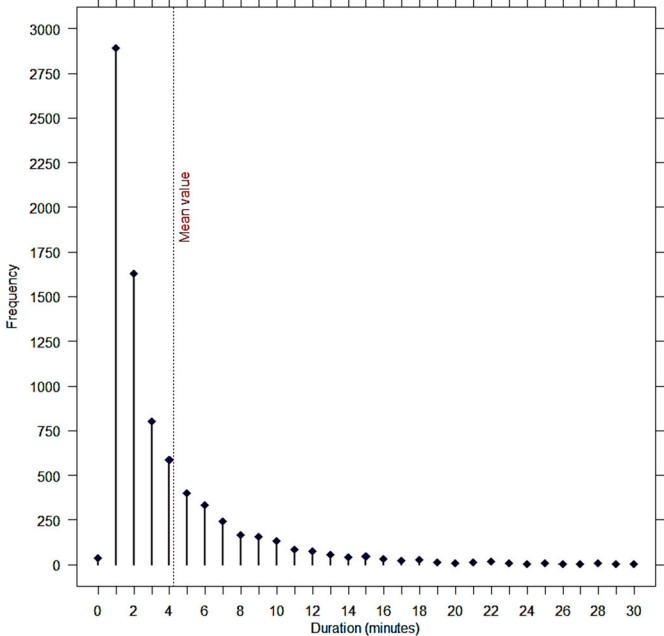

**Figure 6.** Distribution of the operations durations.

To exploit these results, the tasks actually done by the crane must be identified to compare them with expected tasks. Since the data is not sufficient to do this, the durations in value are the only metrics usable. If more time than needed is used for a lifting operation, it means the crane may be overused.

In addition, some tasks need to be understood from the crane point of view. For example, unloading of welded meshes from a delivery truck is theoretically expected to last 90 min. Practically, the welded meshes are taken away by the crane in batches because a truck can deliver more than 18 tons of the item. This weight is impossible to handle with the crane at one time. Moreover, the crane can be called to do some other more important tasks between unloading. This complicates the identification of the tasks because the unloading sequence is cut into smaller operations with various weights depending on the quantity of welded meshes lifted. Hopefully, the location of pick-ups and drop-offs must be the same.

Figure 7 presents the distribution of the weights for each lifting operation. The mean value was 1.82 tons and two-thirds of every operation was below this load. The overall maximum load for the crane was never reached, as indicated by the dataset. The distribution here is quite different from the distribution of the duration. Two peaks were present around 4 and 5 tons, weights corresponding to concrete skips fully loaded or heavy and high formworks.

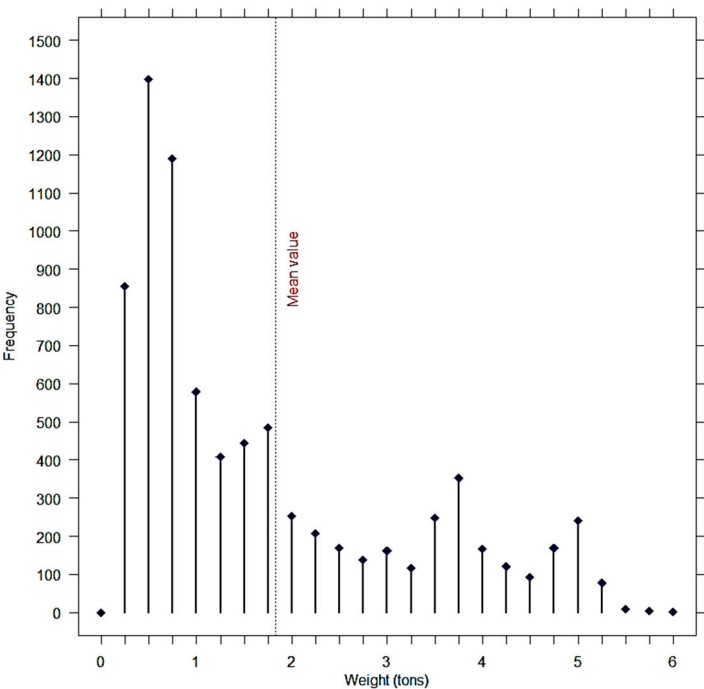

**Figure 7.** Distribution of the operation's weights.

The crane has a permissible load of 10 tons, which is dependent on the trolleying distance. It can happen during a lift that the weight is greater than the allowable load at a particular trolleying distance. For example, a load of 10 tons taken at a trolleying distance of 5 m can only reach 22 m because of the theoretical load curve. If the operator disengages the safety to go further, the crane could do it by undergoing a slowdown or even a stop. Also, it can lead to a safety risk for the crane operator and the workers beneath the hook; a waiting crane is a crane unable to be used and thus, unproductive.

Figure 8 outlines the distribution of distances traveled by the crane obtained by calculating the curvilinear plane abscissa, i.e., the horizontal distance traveled by the hook. The average distance traveled is around 36 m. This indicator is calculated to know if the crane is moving objects at shorter or longer distances. In the second case, it could mean that taking and dropping spots, or areas, were not well managed on the site. For example, taking the concrete skip from one side of the construction site (washing tower) to bring it to the opposite side (concrete delivery area) is time-consuming when it happens every day.

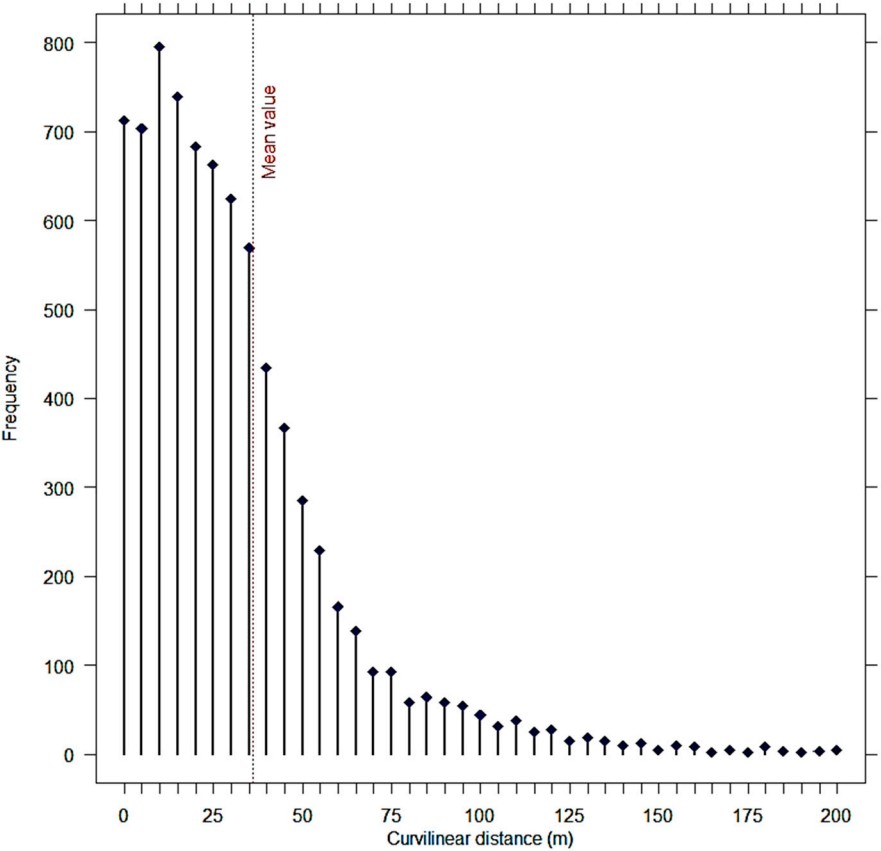

**Figure 8.** Distribution of the curvilinear distance of the operations.

### 4.4. Tower Crane Activities

A common indicator for defining the tower crane's utilization rate was already proposed in the literature [44]. The rate is a percentage between the time spent by the crane working divided by the time the crane is available for work (when the operator is up in the cabin). The "crane working" time is then when the machine is moving and/or when loaded. Therefore, the notion of movement and load is part of the equation.

In this article, the "In motion" state of the crane corresponds to a velocity of the hook greater than zero for more than 5 s. Otherwise, the crane is considered "Stationary". For the "Loaded" state, the data comes directly from the segmented lifting operations because it corresponds directly to the required state (object hanged to the hook). The complementary state is then "Empty". These four states can be associated by pairs which are "In motion when loaded", "In motion when empty", "Stationary when loaded" or "Stationary when empty". Moreover, and for the last state, it is divided into two categories: more than 1 min and less than 1 min stationary time. It is considered that below one minute of immobility, this time is linked to the hook/unhook phase and is not lost.

Figure 9 represents the utilization of the tower crane day by day during the entire construction project. The percentages are calculated through data analysis of daily operations. Daily working hours may vary between 7.11 h (7 h and 7 min) and 12.30 h (12 h and 18 min). The mean value of daily working hours is 9.44 h (9 h and 26 min). If we look closer to the day-by-day utilization levels of the crane, an increase of "Stationary when empty" level was observed at the end of the project. The opposite trend was also visible at the beginning of the project, from mid-June to end-September.

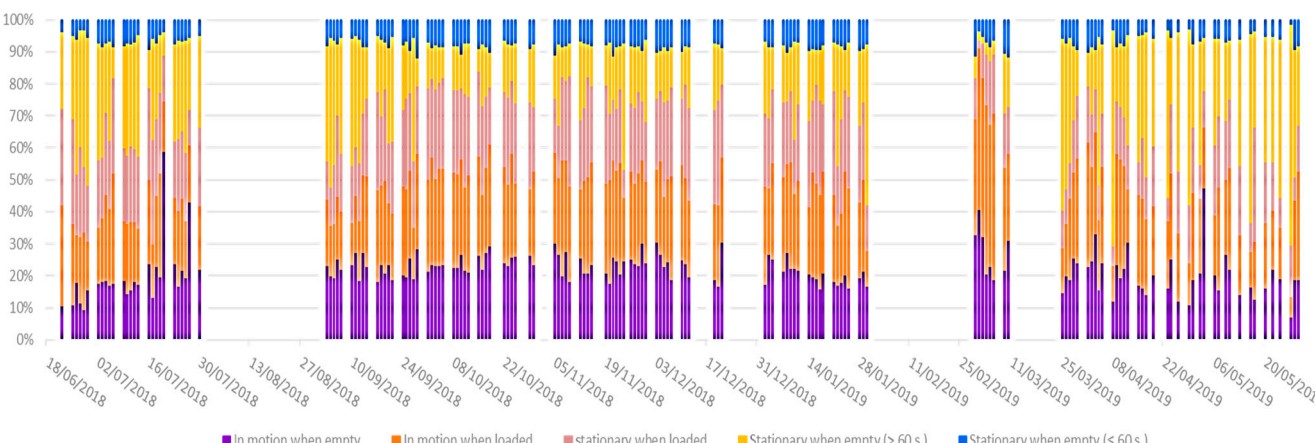

**Figure 9.** Utilization of the tower crane over time.

A steady-state was observable until February 2019, during which percentages did not vary much: this was the cruising speed of the project. The increase in productivity rates could be linked to the presence of a second tower crane operator from the beginning of September 2018. At the end of the presence of the tower crane, the structural phase overlaps with secondary trades. That situation lead to utilization of the tower crane for the deliveries of plasterboards, windows, doors, flooring on the roofs, or on balconies.

Two periods of the long-term failure of the crane datalogger appear in Figure 9, one from 30 July to 3 September and the other from 31 January to 2 March. Moreover, the crane datalogger failed on some days to measure and store the data at the end of the structural phase. As a direct consequence, on 323 days of data logging, 193 days have available data, but only 165 have proper and usable data without measurement issues (one of the measurements is missing during the whole day).

According to Figure 10, 25.89% of the time was used to move objects among the job site locations. Then, percentages were nearly the same when the crane was "Stationary when loaded" (21.48%) and "In motion when empty" (21.62%). The first case was mainly linked to the securing of equipment, such as formworks, shoring towers, scaffoldings, or even prefabricated components (wall, lintel, slabs). The tower crane needed to handle a portion of the load of the equipment while workers install the equipment. The second concerned the trajectories done towards an object to pick up. Finally, 31.02% of the time, the crane was "Stationary when empty". This state was divided into two categories: more than 1 min (23.76%) and less than 1 min (7.26%). It is considered that below one minute of immobility, this time is linked to the hook/unhook phase and is not lost.

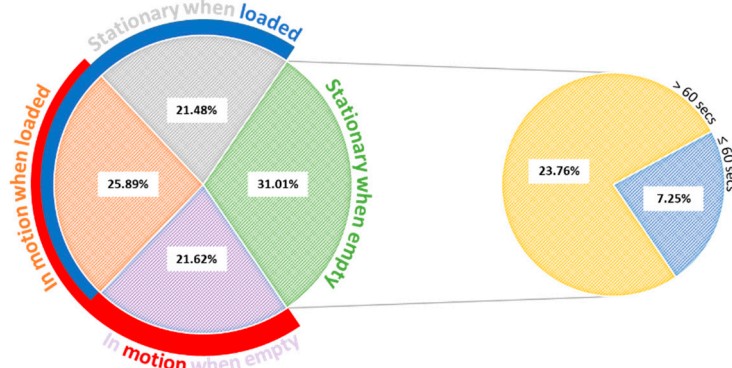

**Figure 10.** Summary of crane utilization during the entire construction project.

In the literature, crane activity indicators exist. To compare our results with the previous work, our "Stationary when empty > 60 secs" becomes "idling", "In motion when

empty" becomes "moving empty" and an aggregate of our "In motion when loaded", "Stationary when loaded" and "Stationary when empty ≤ 60 secs" will correspond to "Lifting/Moving Loaded".

Table 1 illustrates the comparison between the previous studies [45] and the work presented in this article. The measured "Lifting/Moving Loaded" in this article is between 10 and 12% below the others. For the "Moving Empty" state, the value is between 26% and 36% greater than the literature, and the "Idling" state is between 1.5% and 15% higher than the other studies. These results could be explained by the nature of our particular project, which was quite complex because each level had a different layer and custom formworks were needed.

**Table 1.** Average utilization profiles of tower cranes (adapted from * Kumaraswany, 1997).

| SITE (Reference) | Lifting/Moving Loaded (%) | Moving Empty (%) | "Idling" (%) |
|---|---|---|---|
| A (Chan et al., 1995) | 62.3 | 17.1 | 20.6 |
| A * (Leung, 1995) | 62.7 | 16.2 | 21.3 |
| D (Yip, 1997) | 60.7 | 15.9 | 23.4 |
| Present study | 54.63 | 21.62 | 23.76 |

## 5. Conclusions

This article presented a methodology for exploiting data coming from the crane to provide productivity indicators of the construction site. From June 2018 to May 2019, it was possible to study the usage of the crane in a global way over the entire duration of the presence of the crane on the construction site.

The first step of the proposed methodology consisted of collecting the data from the crane data logger, then preparing it by formatting and cleaning the dataset. A visualization step was presented to explain the stakes of the analysis of the crane data to the construction site managers and supervisors.

After that, data were segmented by effective lifting operations using an existing segmentation method. However, the existing method was slightly modified to take into account the hook approach towards the taking point and not the empty motion after the lifting. Projected on a plane, the object's trajectories provided information to the site team on the origin, the destination, and the time the objects were moved.

Moreover, statistical indicators on the operation's duration, typical load, and curvilinear distance of crane movements were performed. For this particular study, the typical lifting operation lasted 4 min and 11 s to move a 1.83 tons object at a distance of 36 m.

Finally, the activities of the crane in terms of load and movement were studied and compared to the existing literature. The crane was "Idling" 23.76% of the time, "Moving Empty" 21.63% of the time, and "Moving Loaded" 54.63%. The last percentage corresponds in principle to productive tasks from the crane's point of view. Comparing the results of this study with the other is not sufficient today to compare the productivity of each project. The sample is larger in our study and the lack of data could have disrupted the statistical distributions of the performance metrics.

## 6. Perspectives

The three statistical distributions can now be used as reference charts for the next similar construction projects. By following our methodology with a data logger installed on another crane becomes possible, and the comparison of the productivity of two or more construction sites.

The data used here was presented in the local coordinate system of the crane, which does not give all the meaning of the displacements of the objects. Further work is in progress to link the construction project's data with the crane data through the exploitation of the BIM model.

**Author Contributions:** T.D. designed the framework of the methodology and carried out the experiment in this case study, which is a part of his PhD thesis work. Z.L. is the thesis director, he supervised the thesis work, including the work in this paper. He also revised the paper on several occasions. A.P. is the co-director of the PhD thesis. He reviewed the present article. S.L. was the project manager responsible for the digitalization of the construction site for Bouygues Construction Company and also revised the paper on several occasions. P.R. is a R&D director for Bouygues Construction Company in the field of Ergonomics and Productivity, and also revised the article. All authors have read and agreed to the published version of the manuscript.

**Funding:** This research was carried out as part of the industrial research chair Construction 4.0, funded by Centrale Lille, Bouygues Construction, The Métropole Européenne de Lille (MEL) and the European Regional Development Fund (ERDF).

**Institutional Review Board Statement:** Not applicable.

**Informed Consent Statement:** Not applicable.

**Data Availability Statement:** Restrictions apply to the availability of these data. Data was obtained from Bouygues Construction and could be available from the authors with the permission of Bouygues Construction.

**Acknowledgments:** We would like to thank the industrial partner Bouygues Construction for financing and supporting the research done in the Construction 4.0 Chair of Centrale Lille. This industrial research chair is also co-financed by the European Metropolis of Lille and the European Union with the European Regional Development Fund.

**Conflicts of Interest:** The authors declare no conflict of interest. The funders had no role in the design of the study; in the collection, analyses, interpretation of data, or in the writing of the manuscript. However, for confidentiality concerns, they acted the decision to publish the results.

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
