# Peer review of "Proposal for Tower Crane Productivity Indicators Based on Data Analysis in the Era of Construction 4.0"

_buildings, doi:10.3390/buildings11010021_

Round 1

Reviewer 1 Report

The article presents a comprehensive analysis of the tower crane’s data during its duty cycle. The subject is very modern in terms of Construction 4.0. The methodology in which the productivity of the construction site on which the tower crane works was measured is described. The individual stages of work are described in details. 

I have reviewed the manuscript and recommended that it can be accepted for publication with minor revisions.

However, some issues need to be expanded or clarified. In particular, the reviewer has several concerns:

1. The Introduction section should be expanded to include at least a few more publications. The article should include a complete background introduction about the current state of the art. Moreover, in References, the Authors cite quite old works. Since the authors deal with a very modern subject, I suggest adding a few publications that describe the subject of Construction 4.0 and/or load motion.

2. The article is correctly written. However, one can find a few errors that do not affect the overall quality of the work. Some example errors: line 89 – Researh objective; line 125 – kilo newton.

3. The style and format of the article should also be improved:

  • Figure captions should be formatted following the Instructions for authors;
  • The title of the article should be capitalized;
  • Fig. 1 should be corrected, especially the font and axis descriptions, to make it more visible;
  • I would suggest enlarging Fig. 2 – then it would be easier to read.
  • Fig. 9a is generally unreadable and confusing;

4. Why, for the purposes of the analysis, the work adopts gravity equal 10m/s2? It is quite a simplification, especially in the case of load transport of such a mass. Authors may refer to some studies or works in which a similar value has been assumed. It should be clarified in the text.

5. The manuscript also does not provide data on sensors and other elements used during the analysis. Providing these data will be a good supplement and will help to better evaluate the obtained results.

Overall the analysis is very impressive! The reviewer hopes the comments will contribute to the improvement of the paper.

Author Response

Point 1: The Introduction section should be expanded to include at least a few more publications. The article should include a complete background introduction about the current state of the art. Moreover, in References, the Authors cite quite old works. Since the authors deal with a very modern subject, I suggest adding a few publications that describe the subject of Construction 4.0 and/or load motion.

Response 1: We added this paragraph in the revised version:

After four industrial revolution, the construction industry has been behind the other industries. The beginning of this decade saw a new concept coming in the industry: The Industry 4.0. The pillars of this concept are Big Data & analytics, Autonomous Robots, Simulation, System Integration, Internet of Things, Cyber security and Cyber Physical Systems, Cloud, Additive Manufacturing and Augmented Reality. All of these axes can be applied to the construction industry. To help with that, an ontology was previously proposed (Dakhli, Danel, and Lafhaj 2019) to use digital tools for the resource management of a construction site. Construction professionals work now together with start-ups and digital companies to bring the Industry 4.0 concepts, digital tools and new way of management into the construction industry. That is the Construction 4.0 (Sawhney, Riley, and Irizarry 2020). Many technological implementations of Construction 4.0 concepts have been published over the last years and now even the workers are 4.0 (Calvetti et al. 2020). We can cite the use of cameras to monitor the worker activities (M. W. Park and Brilakis 2012; Jaselskis et al. 2014; Memarzadeh, Golparvar-Fard, and Niebles 2013) or construction progress, augmented reality tools to enhance the visualization (Mani, Feniosky, and Savarese 2009), or robots for building construction (Bock et al. 2015; Bock 2016; Barnett and Gosselin 2015; Dakhli and Lafhaj 2017).

Point 2: The article is correctly written. However, one can find a few errors that do not affect the overall quality of the work. Some example errors: line 89 – Researh objective; line 125 – kilo newton.

Response 2: These mistakes are now corrected in the revised version.

Point 3: The style and format of the article should also be improved:

  • Figure captions should be formatted following the Instructions for authors;
  • The title of the article should be capitalized;
  • 1 should be corrected, especially the font and axis descriptions, to make it more visible;
  • I would suggest enlarging Fig. 2 – then it would be easier to read.
  • 9a is generally unreadable and confusing;

Response 3: The figures 1 and 2 are corrected in the revised version. Figure 9 is split into 2 different figures to increase the readability of the first part.

Point 4: Why, for the purposes of the analysis, the work adopts gravity equal 10m/s2? It is quite a simplification, especially in the case of load transport of such a mass. Authors may refer to some studies or works in which a similar value has been assumed. It should be clarified in the text.

Response 4: We understand the shift it may cause on the “load” metrics. The equipment used on this tower crane was a prototype of a datalogging system not yet finished and commercialized by the partner. For that reason, we do not really know if the ratio between the stored value and the expected one is due to raw measurement of a load cell (in that case in kN) or due to anything else we do not know. To our knowledge, newer versions of the dat

Point 5: The manuscript also does not provide data on sensors and other elements used during the analysis. Providing these data will be a good supplement and will help to better evaluate the obtained results.

Response 5: We did on-site observations of the crane during full working days. Neither we have installed another load sensor on the crane to check the value of the weight handled by the hook, nor an external equipment to follow the hook movements.

Reviewer 2 Report

This paper mainly uses the data recorded by datalogger installed on the tower crane to analyze the production efficiency of the tower crane, and proposes a new method to calculate the productivity. Datalogger is the inherent program of tower crane, and has high versatility. The proposed calculation method provides a new perspective for the calculation of tower crane productivity, which is more accurate than other methods.

But there are some issues that need to be discussed:

1. How to measure the productivity, compare with what benchmark, and what kind of productivity is reasonable. In practice, the production efficiency can not reach 100%, but how much *% is reasonable?

2.The availability of data recorded by datalogger is less than 50%. The availability of this method is not high.

3. The content of data analysis figure 678 has little effect on productivity calculation.

4 calculate the production rate, how to change the management mode, and achieve the purpose of improving productivity? It is not mentioned in the article.

5. Vision based methods can provide higher accuracy.

Author Response

Point 1: How to measure the productivity, compare with what benchmark, and what kind of productivity is reasonable. In practice, the production efficiency cannot reach 100%, but how much *% is reasonable?

Response 1: The productivity of the crane is a function of crane operator time, actual work done and if this particular work is really productive. We cannot say today that a particular lifting operation is useful or useless and add actual value to the building process, so even if the operator uses the crane at 100%, it can be for doing nothing productive. So, the only indicators available are the one we present in this article.

Point 2: The availability of data recorded by datalogger is less than 50%. The availability of this method is not high.

Response 2: Our case study was focused on one jobsite, which was the first testing ground for the data logging system. This particular system was unfortunately full of issues because it was not yet a commercialized product. The availability and robustness of the system is now way much better (> 99%) and is currently used on other jobsites. Some case studies will be developed and published in 2021.

Point 3: The content of data analysis figure 678 has little effect on productivity calculation.

Response 3: Indeed. They are macroscopic productivity indicators for the crane. The point of these Figures is mainly to generates a kind of abacus to identify the topology of each metric for a particular jobsite. The comparison of the topology and the statistical distributions of many construction sites will then be possible.

Point 4: calculate the production rate, how to change the management mode, and achieve the purpose of improving productivity? It is not mentioned in the article.

Response 4: The data to calculate the production rate of the construction site are not all yet available (actual worked hours of the workers for example). A way for improving productivity for the cranes could be taking less time to do the tasks. In reality, comparing the expected duration with the actual one is more suitable. Moreover, if we consider that taking more load at each pick-up is a way to dispatch more “work” around the site, it could be also an improvement.

Point 5: Vision based methods can provide higher accuracy.

Response 5: We totally agree with that but because of the GDPR Policy in Europe and our industrial partner, the vision-based methods have not been retained. Of course, cameras are cheap equipment to collect data but the overall infrastructure to transmit, store and process the visual flow is expensive and does not offer a quick return-on-investment for the industrial partner. Data governance is also a “hot topic” for each company now.

Reviewer 3 Report

This paper presents a good study of crane efficiency but it is not indicative of overall construction productivity as implied in the title and on lines 91-92. The crane efficiency measurement requires the addition of some benchmark references to measure against. I suggest including and accounting for logistical plans as they can greatly alter how the crane is used which would affect productivity. Another factor to consider is scheduling deliveries to not alter crane operations once a task has begun (Lines 260-261), an argument could be made that this is a loss of crane efficiency due to mismanagement of the crane operations and not indicative of overall project productivity.

Title: Adjusting the title to refer to crane efficiency instead of construction site productivity would improve relevancy to the content.

Lines 21-22: Please provide more recent refences if possible

Line 23: Recommend rephrasing to "Is it the whole industry or could this assumption be nuanced?

Line 29: Please provide more recent references if possible

Lines 35-41: I recommend looking up more references, and elaborating on the different techniques that exist for productivity measurement such as the measured mile approach.

Lines 91-92: "It could imply that the low productivity of the lifting machine on-site could be one of the reasons for the low productivity of the construction sector" - Please provide a reference or elaborate further on the connection between crane efficiency and overall construction productivity

Lines 167-168: Minor grammatical errors, some rephrasing required

Line 189: Were the authors able to break out the "weather vane" portion that occurred during work hours, if any ?

Figure 2: It would be beneficial to overlay this figure on a site logistics plan, this may offer other topics of discussion

Lines 255-256: Please elaborate further on the relation of crane overuse and construction delays, would help to provide some data or references for this statement. 

Author Response

Point 1: The crane efficiency measurement requires the addition of some benchmark references to measure against. I suggest including and accounting for logistical plans as they can greatly alter how the crane is used which would affect productivity.

Response 1: I totally understand the idea of including the logistical plan in our study. Unfortunately, our industrial partner and the French construction industry in general does not have this kind of plan for the crane activities. The best we have observed on site about crane activity planning is that the foreman discusses with the crane operator at the beginning of his shift on what they are planning to do in the day, but in a general way, not in details. Some recommendations have been formulated to our partner to provide a better sequence of operations to the crane operator.

Point 2: Another factor to consider is scheduling deliveries to not alter crane operations once a task has begun (Lines 260-261), an argument could be made that this is a loss of crane efficiency due to mismanagement of the crane operations and not indicative of overall project productivity.

Response 2: We agree with you concerning the mismanagement of the crane operations and the issues it could cause on the productivity. It is often the case on site when the concrete truck arrives and the crane has to be ready to handle it, so do nothing else, potentially important. Although, the crane will do the job anyway, but later. So, in the worst case, the workers will wait the crane, inducing a loss a productivity for their side, while not affecting the crane efficiency.

Point 3: Title: Adjusting the title to refer to crane efficiency instead of construction site productivity would improve relevancy to the content.

Response 3: We changed the title of the article to be more coherent with the work done. We propose the following: “Proposal for tower crane productivity indicators based on data analysis in the era of construction 4.0”

Point 4: Lines 21-22: Please provide more recent refences if possible

Response 4: To our knowledge, there is no such reference that study the whole construction industry and its productivity. We can however cite the lasts report of McKinsey “The next normal in construction: How disruption is reshaping the world’s largest ecosystem” or the Mark Farmer’s “Modernise or Die” report.

Point 5: Line 23: Recommend rephrasing to "Is it the whole industry or could this assumption be nuanced?

Response 5: We replaced the sentence by “But the construction industry is fragmented: building houses, apartments of offices is not the same than digging tunnels or building bridges. The question is then: are these branches are all suffering of this low productivity rate or one branch is suffering more than the others?”

Point 6: Line 29: Please provide more recent references if possible

Response 6: We added (Costin et al., 2015) and (Mohtat, 2016) as references for worker’s efficiency and material tracking.

Point 7: Lines 35-41: I recommend looking up more references, and elaborating on the different techniques that exist for productivity measurement such as the measured mile approach.

Response 7: Thank you for sharing the “measured mile approach” method. We added a paragraph on the available methods and references:

Among them, we can cite the measured mile approach [Ibbs and Liu, 2005], [Ibbs, 2012], many techniques of motion and time study such as work or activity sampling [Handa and Abdalla, 1989], [Luo et al., 2018], stopwatch study [Taylor, 1919], photography, videotaping, time lapse video [Teizer & Vela, 2009], [Yang et al., 2014], questionnaire surveys or even automated productivity measurement system [Kim et al. 2011].

Point 8: Lines 91-92: "It could imply that the low productivity of the lifting machine on-site could be one of the reasons for the low productivity of the construction sector" - Please provide a reference or elaborate further on the connection between crane efficiency and overall construction productivity

Response 8: This is a formulated hypothesis that we were trying to answer. But for now, the low productivity of the crane appears to be linked to the construction site organization (deliveries, management of workers). We deleted it to not lose the reader.

Point 9: Lines 167-168: Minor grammatical errors, some rephrasing required

Response 9: The sentence is modified with the following:

The segrentation method gives the pick-up and the drop-off time of each lifted object. The duration of the lifting operation is then deduced from the calculus: drop-off time – pick-up time = operation duration.

Point 10: Line 189: Were the authors able to break out the "weather vane" portion that occurred during work hours, if any?

Response 10: We don’t fully understand the question but we will try to answer. The “weather vane” mode of operation is activated when the crane operator is not up in the cabin. So, when it is activated, it is not considered as “work hours”. Moreover, this time is not considered for the productivity measurement.

Point 11: Figure 2: It would be beneficial to overlay this figure on a site logistics plan, this may offer other topics of discussion

Response 11: The site logistics plan could indeed bring the meaning of the lifting operation and can be used to know where an object was taken then dropped off. In fact, to exploit the data with a business point of view, we could also link the site logistics plan, the BIM model and theoretical sequence of operations. But this particular article is focused on providing macroscopic indicators without the construction project documents and the relative knowledge of the construction site environment, only the crane data. Moreover, adding too much information on the graphic could lead to the decrease of the understanding of the work done here.

Point 12: Lines 255-256: Please elaborate further on the relation of crane overuse and construction delays, would help to provide some data or references for this statement.

Response 12: This is an interesting question. First, the theoretical planning of operations is realized by considering many parameters such as the method of construction (prefabrication or cast-in-place), the equipment on site (number of formworks), the linear distance of walls and the surface of the floors. The number of cranes on site is also a factor because you can have interferences between them. So, each task takes time, and the sum of all the task’s time gives the time you need to build with the theoretical parameters. If each task takes more time than expected, the construction project duration will mathematically increase.

Then, I understand this delay could be managed by doing more hours per day to compensate it. The consequence is that the cost of the project will be affected because of the paying of overtime hours. To my little experience on construction sites, when there are delays, the first thing the construction site manager look at is the crane utilization rate. So, we will delete our hypothesis from the article and work on it to provide better answers.